# A Systematic Review of Fintech and Banking Profitability

Adey Tarawneh [1,2,*], Aisyah Abdul-Rahman [1,3], Syajarul Imna Mohd Amin [1] and Mohd Fahmi Ghazali [1]

[1] Faculty of Economics & Management, Universiti Kebangsaan Malaysia (UKM), Bangi 43600, Selangor, Malaysia; eychah@ukm.edu.my (A.A.-R.); imna@ukm.edu.my (S.I.M.A.); fahmi@ukm.edu.my (M.F.G.)

[2] Department of Management, Ahmad Bin Mohammed Military College, Doha 22988, Qatar

[3] Institute of Islam Hadhari, Universiti Kebangsaan Malaysia (UKM), Bangi 43600, Selangor, Malaysia

[*] Correspondence: p111582@siswa.ukm.edu.my or tarawneh.oday@abmmc.edu.qa; Tel.: +974-33921744

**Abstract:** Financial technology (Fintech), characterized as technology-driven financial innovation, has catalyzed significant economic growth across various nations. The Fintech sector has experienced remarkable expansion, boasting vast user numbers. While regions like the United States and China have seen accelerated Fintech development, other areas like Western Europe, Eastern Asia, and the Middle East continue their evolutionary journey with this technology. Our research offers a systematic review of contemporary literature, probing the crucial Fintech metrics affecting bank profitability and identifying the primary factors influencing these profits. This review introduces a holistic methodology for quantitatively assessing the evolving Fintech measures and their interplay with determinants of bank profitability. According to the Preferred Reporting Items for Systematic Reviews and Meta-Analyses (PRISMA) guidelines, our study evaluates 28 articles from Web of Science and Scopus databases from August 2019 to August 2023. Findings delineate two principal themes: Fintech measures at both bank and country levels and determinants of profitability, encompassing bank-specific and country-specific variables. We utilize the Theories, Constructs, Contexts, and Methods framework to chart the course for future research. Our insights bear significance for theoretical progression and practical implementation, offering academics, banking professionals, and policymakers a nuanced comprehension of the nexus between Fintech and bank profitability.

**Keywords:** Fintech; banks; profitability; determinants; PRISMA

## 1. Introduction

The banking sector plays a pivotal role in a nation's economic development. An efficient banking system underpins the efficient allocation and management of funds and financial investments, fortifying the nation's economic architecture (Isayas 2022). On the other hand, instability within the banking sector can adversely affect economic expansion (Safiullah and Paramati 2022). Banks should examine their profitability performance to enhance economic growth as it contributes positively to overall economic development. The global financial crisis of 2008 and a significant decline in interest rates adversely impacted bank profitability. This circumstance prompted new concerns for policymakers to focus on bank profitability as a critical consideration (Klein and Weill 2022). Profitability is essential in banking for sustainability and providing returns to investors, emphasizing the need for banks to focus on their financial performance to contribute effectively to the economy (Isayas 2022).

The factors influencing bank profitability can be categorized into micro and macro factors. Bank size is among the most popular micro factors (Petria et al. 2015). The size of a bank holds significance, as larger banks are expected to benefit from economies of scale, resulting in a potential advantage in terms of costs or profitability (Katsiampa et al. 2022). Besides size, Liquidity can impact banks' profitability, as those with higher levels of Liquidity are more likely to exhibit superior performance (Nguyen et al. 2021). In addition, loan quality affects bank profitability, as high-quality loans ensure consistent

interest income and require lower provisions for losses, increasing revenue and reducing operating costs (Adalessossi 2023).

Conversely, portfolios with poor-quality loans require more enormous loss reserves and higher regulatory capital, reducing profitability due to increased risk and operating expenses. Another factor affecting bank profitability is the cost-to-income ratio, a key indicator of operational efficiency. A higher ratio indicates higher efficiency and profitability, as it suggests the bank is managing its operating expenses well relative to its income. In contrast, a higher ratio means lower profitability due to higher costs in income (Haddad and Hornuf 2022).

Concerning macro factors, banks' profitability can be influenced mainly by GDP. Economic growth stimulates increased lending by banks, allowing them to impose higher margins. Consequentially, this contributes to an enhancement in the quality of their assets (X. Wang et al. 2021). Additionally, Inflation can impact the profitability of banks. When Inflation is unexpected and banks do not adjust their interest rates accordingly, costs may rise more rapidly than revenues, negatively impacting bank profits (Phan et al. 2020). Moreover, the concentration level in the banking sector is a critical macroeconomic variable that greatly influences bank profitability (X. Wang et al. 2021). Higher concentration indicates that fewer banks are dominating the market, which can lead to less competitive pressure, allowing these banks to charge higher fees and interest rates, thus increasing their profit margins. Conversely, in a market with a lower concentration and more competition, banks may face challenges in maintaining high levels of profitability due to tighter margins and greater competitiveness (Nguyen et al. 2021; X. Wang et al. 2021)

In the face of emergent technologies, the banking landscape is witnessing intensified competition, compelling institutions to redouble their efforts in optimizing performance to maintain a competitive edge (Murinde et al. 2022). After the global financial crisis 2008, Fintech rose to prominence, with individuals increasingly trusting these techRnological and economic innovations (Rabbani et al. 2022). Fintech, representing the convergence of financial services and technology, has materialized as a revolutionary force in today's era, catalyzing innovation and propelling economic growth. Its significance stems from its capability to challenge established financial paradigms, broaden the accessibility of financial services, and stimulate entrepreneurship and innovation across diverse sectors (Hoang et al. 2022; Sun et al. 2023). A significant facet underscoring Fintech's role in fostering innovation and economic expansion is its innate potential to augment financial inclusion. Without technological advancements, Fintech enterprises have pioneered solutions enabling previously marginalized populations, especially in developing regions, to have access to fundamental financial services. Innovations like mobile payment gateways, digital wallets, and peer-to-peer lending mechanisms have facilitated broader participation in the formal financial structure, equipping individuals and enterprises with sophisticated tools for savings, investments, and astute financial management. Such inclusive practices inherently amplify economic vibrancy and growth as an expanding population gains entry into the financial continuum (Guang-Wen and Siddik 2023; Alaassar et al. 2023).

Fintech has fundamentally reshaped the operational dynamics of banks. Conventional banking infrastructures often entail intricate procedures and stringent benchmarks for credit accessibility, stymieing the ascendancy of budding startups and inventive enterprises. In contrast, Fintech platforms harness diverse financing avenues and adopt avant-garde risk evaluation frameworks, facilitating expedited and precision-based credit adjudications (Sidaoui et al. 2022). This agile facilitation of capital ensures that banks can channel investments into research and innovation, broaden their operational horizons, and introduce groundbreaking products and services. By fostering innovation, Fintech emerges not merely as a technological marvel but as a potent stimulant for economic progression, employment generation, and heightened competitive advantage (E. Li et al. 2023).

Fintech has become increasingly crucial for innovation in the banking sector, revolutionizing traditional banking practices and driving significant transformations. Its impact can be seen in various areas, including profitability performance, customer experience,

operational efficiency, risk management, and product development (Chen et al. 2021). One of the critical aspects of Fintech's importance in the banking sector is its ability to enhance the customer experience. Transitional banking typically encompasses extensive procedures, paperwork, and restricted access. Fintech innovations have transformed this scenario by offering digital banking platforms, mobile apps, and online payment systems, which deliver convenience, swiftness, and tailored services to clients. Fintech enables customers to conveniently manage their accounts, transfer funds, execute payments, and even apply for loans or investments from the convenience of their own devices. This enhanced customer experience meets consumer expectations and lures new clientele, leading to heightened customer retention and bank customer acquisition (Riikkinen and Pihlajamaa 2022).

Fintech has substantially contributed to enhancing operational efficiency in the banking industry (R. Wang et al. 2021). Through automation, artificial intelligence, and data analytics, Fintech solutions have rationalized operations, minimized manual errors, and optimized resource utilization (Ghandour 2021). Robotic Process Automation (RPA) and machine learning algorithms are more efficient and precise than humans in handling repetitive tasks such as data entry or document verification (Villar and Khan 2021). Furthermore, data analytics tools empower banks to glean valuable insights into customer behavior, detect fraud, and assess risk, facilitating proactive decision making. These technological advancements enhance operational efficiency, reduce costs, and liberate human resources to focus on more intricate and value-added responsibilities (Murinde et al. 2022).

Additionally, Fintech has played a pivotal role in revolutionizing risk management procedures within the banking sector (Mitra and Karathanasopoulos 2020). With the rising volume and complexity of financial transactions, banks require advanced tools to evaluate and address risks effectively. Fintech solutions furnish real-time monitoring, predictive analytics, and early warning systems for identifying potential risks and vulnerabilities (Yao and Song 2021). By leveraging big data and machine learning algorithms, banks can detect fraudulent activities, assess creditworthiness, and manage market and operational risks more efficiently. This enhanced risk management capability minimizes financial losses, ensures regulatory compliance, and strengthens the banking system's stability, ultimately contributing to economic growth (Milojević and Redzepagic 2021).

In addition to improving the bank's operational efficiency, Fintech has spurred innovation in product development and expanded the range of services banks offer. Traditional banks often had limited product portfolios and faced challenges adapting to changing customer demands (Al-Matari et al. 2022). Fintech startups, however, have introduced innovative financial products and services, such as peer-to-peer lending, robo-advisors, crowdfunding platforms, and digital wallets (Kumar et al. 2020). These offerings cater to evolving customer needs, provide alternative financing options, and promote financial inclusion (Kumar et al. 2020). Traditional banks have responded to this innovation by collaborating with Fintech firms, investing in their digital transformation, or developing their in-house Fintech capabilities (Ye et al. 2022). This collaboration between Fintech and banks drives product innovation, improves competitiveness, and stimulates economic growth in the banking sector (Hornuf et al. 2021).

While there have been studies attempting systematic reviews of Fintech measures, these studies have yet to offer a comprehensive perspective on the entirety of Fintech measures and their impact on banking profitability. Recent studies focus mainly on specific Fintech innovations such as Peer-to-Peer lending (P2P), Third-Party Payment (TPP), and Crowdfunding (Rabbani et al. 2022; Alhammad et al. 2021) while also exploring the potential advantages or challenges that Fintech companies may introduce to the banking sector (Elsaid 2021). On the other hand, this study aims to fill the existing gap in the literature by applying a comprehensive assessment of Fintech measures at both the bank level (such as digitalization, ATM ratio, and E-payment) and country level (such as P2P lending, TPP, and Crowdfunding). It illuminates the relationship between these measures and bank profitability. Moreover, this SLR lightens the other vital factors that commonly influence profitability in the Fintech era.

Additionally, this research utilizes innovative methodologies: thematic analysis, as delineated by Page et al. (2021), to identify and explain the most significant predictors adopted, and the Theories, Constructs, Contexts, and Methods (TCCMs) framework proposed by Chen et al. (2021), to outline a prospective research agenda. In short, this study addresses the deficiencies in prior research by evaluating how different significant Fintech measures influence banking profitability. This review is anchored on a pivotal research question: What Fintech measures have been employed in previous studies, and how do they affect bank profitability? To expand on this, our study aims to address a set of specific research questions.

1. What are the most commonly utilized Fintech measures in the literature and their impact on bank profitability?
2. What are the other determinants affecting bank profitability in the Fintech era?

This paper is structured into several sections. Section 2 outlines the process, while Section 3 summarizes the results and discussion, encompassing content analyses. Section 4 scrutinizes knowledge gaps and potential research topics. Finally, Section 5 concludes the study.

## 2. Methodology

Following Khatib et al. (2023), this study opted for a systematic review due to its efficacy in thoroughly investigating a specific field. The Systematic Literature Review (SLR) technique is widely employed in management, finance, and economics. Another benefit of utilizing SLR is its ability to minimize subjective and biased conclusions, enhancing the investigatory nature of the discussion by mitigating academic preferences in sample material selection (Khatib et al. 2023).

### 2.1. The Review Protocol (PRISMA)

This research was directed by the recommended guidelines for reporting systematic reviews and thematic analysis, known as the Preferred Reporting Items for Systematic Reviews and Thematic Analysis (PRISMA) (Page et al. 2021). According to Sierra-Correa and Kintz (2015), it presents three distinct benefits: (1) it establishes a well-defined research query to facilitate thematic examination of the system, (2) it outlines criteria for including and excluding content, and (3) it strives for verification. The SLR process commenced with formulating appropriate review inquiries in line with PRISMA's guidance. A three-phase document search methodology was created and executed, encompassing identification, screening, and eligibility stages.

### 2.2. Development of Research Question

The research question for this study was formulated using the PICO framework, which stands for Population, Interventions, Comparators, and Outcomes. PICO is a helpful tool for authors to create pertinent research questions for a review (Lockwood et al. 2015). Consequentially, we integrated three fundamental elements into the review: Banks (representing the population), Fintech measures and profitability determinants (as the focus of interest). It is important to note that the context of Fintech measures was based on their application within banks and at the country level. These three components guided us in shaping a central research question: What are the most commonly utilized Fintech measures in conjunction with other determinants affecting bank profitability?

### 2.3. Systematic Search Strategies

The process of systematic search strategy comprises three primary phases: identification, screening, and determining eligibility.

#### 2.3.1. Identification

Identification involves seeking synonyms, correlated terms, and different iterations of the primary keywords used in the study: "Fintech" and "banking industry". This approach

broadens the scope of database exploration, uncovering additional pertinent articles for the review. The selection of keywords was guided by (Okoli 2015) and sourced from various outlets, including an online thesaurus, keywords utilized in prior research, suggestions from Scopus, and input from subject experts. We refined the established keywords for WoS and Scopus, then proceeded to craft a search string (employing techniques such as Boolean operators, phrase searches, truncation, wildcards, and field codes), as depicted in Table 1.

**Table 1.** Search strings and databases.

| Database | Search Strings |
|---|---|
| Web of Science (WoS) | (("Fintech" OR "Financial Technology" OR "Digital Bank*" OR "Digitalization" OR "Crowdfunding" OR "P2P Lending" OR "Peer to Peer Lending" OR "Shadow Bank*" OR "Online Bank*" OR "Mobile Money" OR "Mobile Transaction*" OR "Payment System*") AND ("Bank*" OR "Financial Institut*") AND ("Performance" OR "Profitability")) |
| Scopus | (("Fintech" OR "Financial Technology" OR "Digital Bank*" OR "Digitalization" OR "Crowdfunding" OR "P2P Lending" OR "Peer to Peer Lending" OR "Shadow Bank*" OR "Online Bank*" OR "Mobile Money" OR "Mobile Transaction*" OR "Payment System*") AND ("Bank*" OR "Financial Institut*") AND ("Performance" OR "Profitability")) |

Note: All the asterisk above where the keywords used during the search process. Digital Bank*" = Digital bank OR digital banks, Shadow Bank*" = Shadow bank OR Shadow banks, Online Bank*" = Online bank OR online banks, Mobile Transaction* = Mobile transaction OR mobile transactions, Payment System*" = Payment system OR payment systems, Bank* = Bank OR banks "Financial Institut*" = Financial insitiution OR financial insitiutions.

### 2.3.2. Screening

We downloaded the identified papers using Mendeley software (version is: 1.19.8.0) and exported them to Excel. Subsequently, we employed an automated criteria selection process, utilizing the sorting functions of the WoS and Scopus databases (as per Okoli 2015), to sift through all 1808 articles before conducting an in-depth review. Meanwhile, the search was confined to August 2023, building upon the search process initiated in June 2021. This rationale led to the selection of the period from August 2019 to August 2023 as one of the inclusion criteria. This limitation of the last five years aimed to capture the most up-to-date research and identify the latest advancements in this continuously evolving sector.

Regarding the timeframe, a span of five years was deemed adequate for observing research trends and associated publications in the field of Fintech (Shaffril et al. 2021). Furthermore, we deliberately omitted the term "Financial Innovation" from our search query to focus exclusively on articles related to Fintech. To eliminate any potential ambiguities, we incorporated research within our review with concrete data published in English.

During this procedure, 46 duplicate items and 1422 articles that did not adhere to the inclusion criteria were excluded from the review process (Table 2). The remaining 322 articles underwent the third step, assessing their eligibility.

**Table 2.** Inclusion and exclusion criteria.

| Criteria | Eligibility | Exclusion |
|---|---|---|
| Type of Literature | Indexed journals containing research articles that only include empirical data which emphasize on quantitative studies | Journals not included in indexing, systematic literature review publications, book chapters, conference proceedings, and papers focused on conceptual discussions |
| Language | Only English articles | Non-English articles |
| Timeline | August 2019–August 2023 | <2019 |
| Unit of analysis | Fintech and bank profitability | Non-banking institutions, and non-profitability and Fintech papers |

### 2.3.3. Eligibility

During the eligibility step, we conducted a manual review of the retrieved articles to confirm that they met the necessary criteria, following the initial screening process. This manual review involved examining the titles and abstracts of the articles. Among these, 265 articles were excluded because they predominantly focused on qualitative and quantitative evaluations of customer trust, customer perception, and manager perceptions related to Fintech adoption and the performance of Fintech companies. The articles did not align with our main objective for this study, which is to include quantitative articles explicitly focusing on Fintech and bank profitability.

### 2.3.4. Data Extraction

Referring to Figure 1, we examined the 28 selected articles and extracted the necessary data. A thorough examination of each publication was conducted with a particular focus on the abstracts, findings, and discussions. Pertinent information was gathered following the research questions, and studies that addressed these inquiries were chosen and structured within a table. We conducted thematic analyses to achieve this, subsequently identifying themes and subthemes through pattern recognition, grouping, and counting techniques (Braun and Clarke 2006).

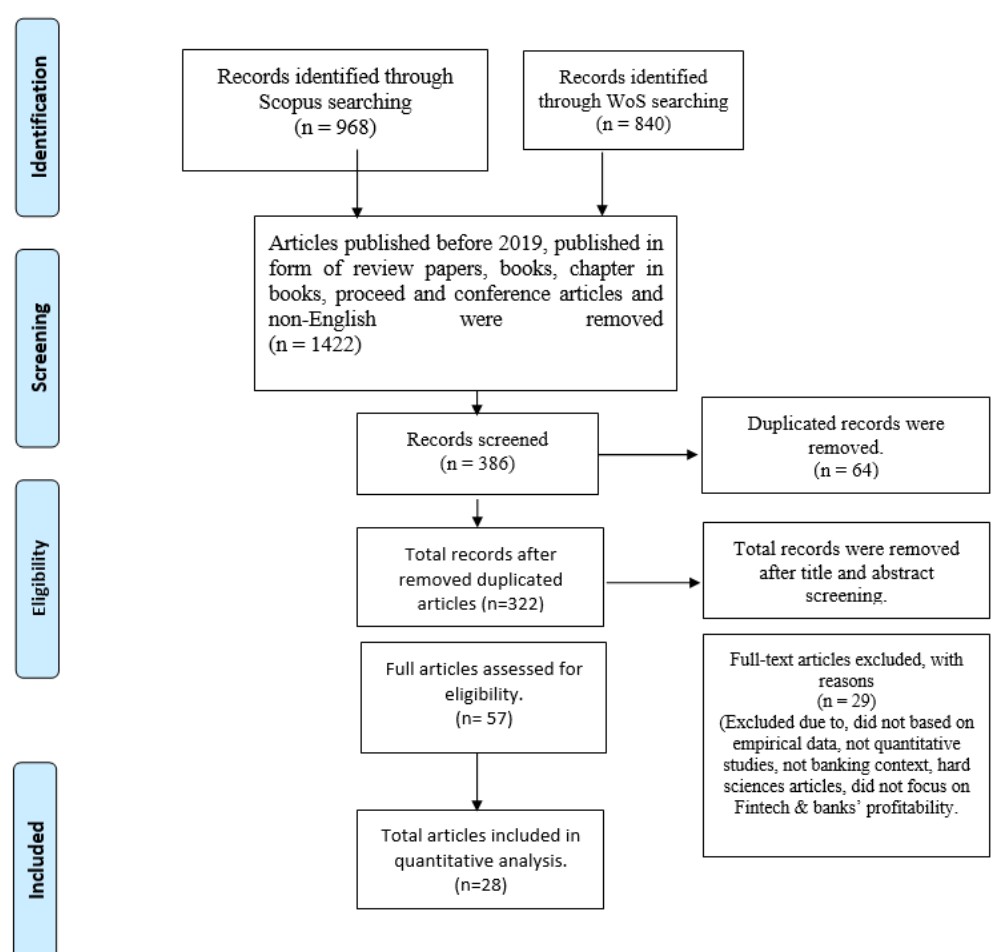

**Figure 1.** PRISMA study flow diagram. Source: (Page et al. 2021).

## 3. Results and Discussion

According to the review, various theories regarding Fintech's impact on bank profitability were employed in research, as summarized in Table 3 and Figure 2. Among these theories, the Consumer and Disruptive Innovation theories emerged as the most favored

approaches in Fintech and bank profitability research. These studies utilized an extended version of these theories, which acknowledged that Fintech companies were altering the landscape of bank profitability by supplanting traditional services offered by established banks. Moreover, the Disruptive Innovation Theory posits that new entrants, such as Fintech firms, harness innovative technology to provide more accessible and cost-effective goods and services, thereby generating competition within the market. Other preferred theories included Solow's Paradox Theory and the Agency Theory. In general, individual articles utilized the Financial Innovation Theory, the Structure–Conduct–Performance Theory, and the Transaction Cost Theory to investigate how Fintech measures impact banks' profitability.

**Table 3.** Theories are used in the research on Fintech and bank profitability.

| Theory | Frequency |
|---|---|
| Consumer Theory | 4 |
| Disruptive Innovation Theory | 4 |
| Solow's Paradox Theory | 2 |
| Agency Theory | 2 |
| Financial Innovation Theory | 1 |
| Structure–Conduct–Performance Theory | 1 |
| Transaction Cost Theory | 1 |

Note: Some reviewed articles did not explicitly anchor their findings within established theoretical frameworks.

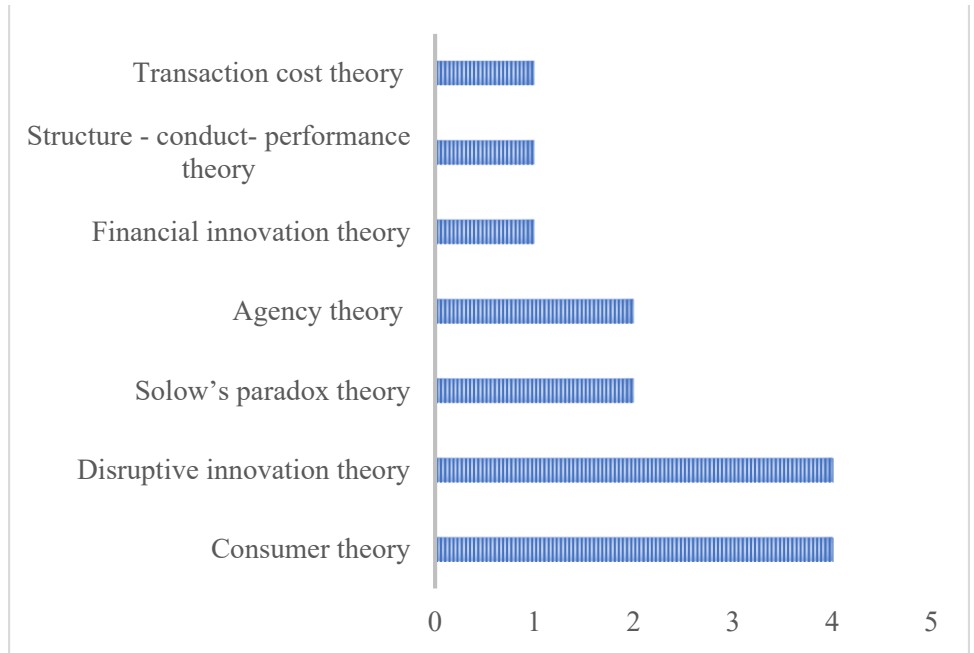

**Figure 2.** The theories used in the research on Fintech and bank profitability.

Among the collected articles, 12 research studies were carried out across different countries (cross-countries), with seven conducted in China and three in Indonesia. In contrast, single studies were conducted in the USA, Malaysia, Jordan, Palestine, Saudi Arabia, and Zimbabwe, as summarized in Table 4 and Figure 3. On the other hand, the frequently employed analytical approaches encompassed both linear and non-linear methods. Linear and non-linear regression are statistical techniques used for modeling the connection between independent variables (predictors) and a dependent variable (Kumar 2006). Linear regression is applied when the relationship among variables seems linear or can be approximated as a straight line. In contrast, non-linear regression is employed when the relationship between variables is non-linear and cannot be effectively represented

by a straight line (Kumar 2006). Among the gathered articles, the majority utilized linear regression analysis (Adalessossi 2023; Kharrat et al. 2023; E. Li et al. 2023; Yudaruddin 2023; Aguegboh et al. 2022; Al-Matari et al. 2022; Haddad and Hornuf 2022; Katsiampa et al. 2022; Almulla and Aljughaiman 2021; Awwad 2021; Bashayreh and Wadi 2021; Carlini et al. 2021; Del Gaudio et al. 2021; Nguyen et al. 2021; X. Wang et al. 2021; Wu and Yuan 2021; Chen et al. 2020; Dong et al. 2020; Forcadell et al. 2020; Phan et al. 2020; Tobing and Wijaya 2020; Wadesango and Magaya 2020; Han et al. 2019), while others opted for non-linear regression (Ben Bouheni et al. 2023; L. Li et al. 2023; Zhao et al. 2022). A solitary paper used mixed methods to combine linear and non-linear approaches (Lu 2022; Chhaidar et al. 2022) (Table 5 and Figure 4).

**Table 4.** The contexts explored in research concerning Fintech and its impact on bank profitability.

| Country | Frequency |
|---|---|
| Cross-country | 12 |
| China | 7 |
| Indonesia | 3 |
| Malaysia | 1 |
| Zimbabwe | 1 |
| USA | 1 |
| Jordan | 1 |
| KSA | 1 |
| Palestine | 1 |
| Total: | 28 |

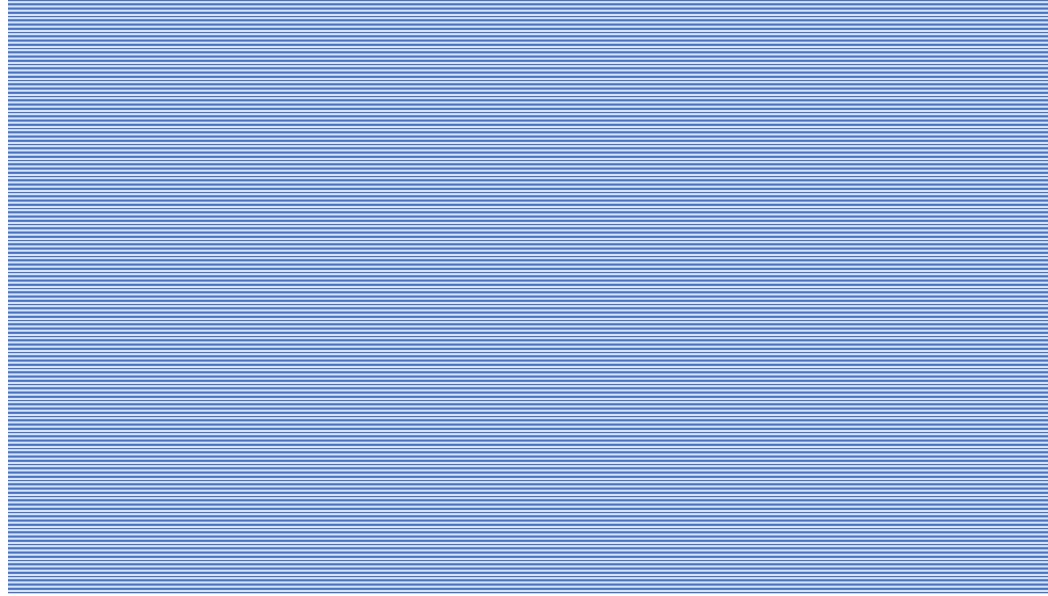

**Figure 3.** The contexts explored in research concerning Fintech and its impact on bank profitability.

**Table 5.** Methods of analysis used in the influence of Fintech and bank profitability.

| Method | Frequency |
|---|---|
| Linear | 23 |
| Non-Linear | 3 |
| Mixed | 2 |

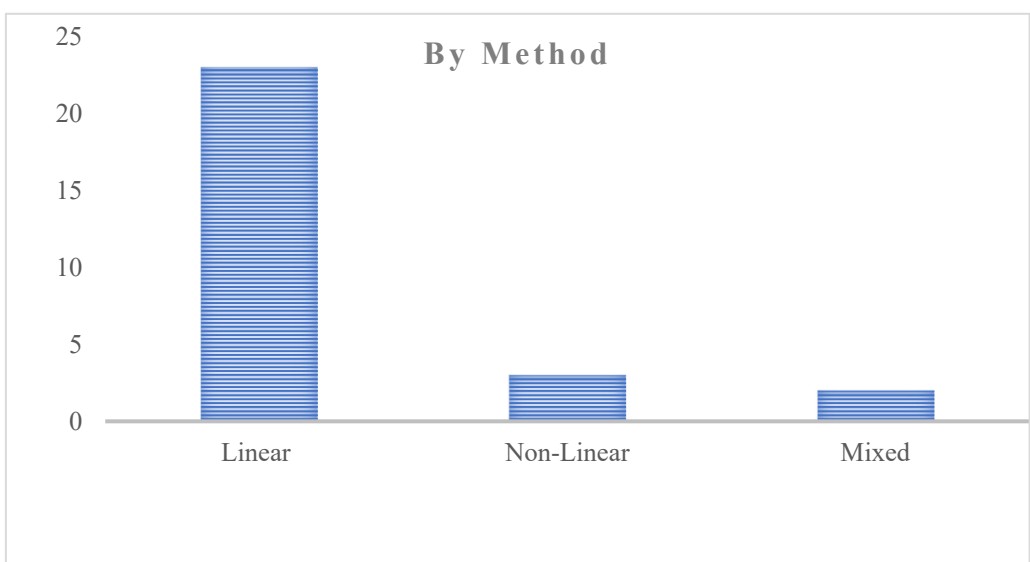

**Figure 4.** Methods of analysis used in the impact of Fintech and bank profitability.

Our examination revealed that specific Fintech metrics were initially associated with country-level data, including Money Transfer (MT), Mobile Banking (MB), and the ATM ratio. A closer scrutiny of the articles revealed that these metrics were also operational at the bank level. Furthermore, concerning Fintech services offered by banks, certain studies either conceptualized these variables as dummy variables or utilized keyword filtering to pinpoint Fintech-centric content. Concerning country-level Fintech indicators, it was evident that several articles encompassed diverse Fintech entities, such as Peer-to-Peer (P2P) platforms and third-party payment systems. Consequently, we bifurcated Fintech metrics into two predominant categories: Bank Level and Country Level.

Within the category of bank-level measures, we delineated seven distinct subthemes. We have articulated two subthemes for the country-level category: Fintech Firms and Information and Communication Technology (ICT). We discerned five additional subthemes in the "Fintech Firms" subtheme. Adopting a methodology akin to our treatment of Fintech metrics, we employed a similar stratification for other determinants of profitability. This approach culminated in identifying two overarching themes: Bank-Specific Variables and Country-Specific Variables. The former encompassed eight subthemes, whereas the latter comprised three. Our analysis yielded four primary themes and a comprehensive set of 23 subthemes.

Table 6 offers a comprehensive synthesis of the Fintech metrics utilized in investigating the nexus between Fintech and bank profitability. Within the bank-level ambit, Digitalization, ATM ratios, and E-payments were salient themes, represented in 5, 4, and 4 studies, respectively. In juxtaposition, from the country-level vantage, dominant themes were peer-to-peer, third-party payments, and Crowdfunding, which accounted for 11, 9, and 8 studies, respectively. Delving into alternative determinants of profitability, our observation discerned that under bank-specific variables, Size, Nonperforming Loans (NPLs), and Liquidity were the predominant subthemes, featured in 13, 5, and 5 studies, respectively. Conversely, from the country-level perspective, GDP and Inflation emerged as leading variables, documented in 9 and 8 studies, respectively. Subsequent sections undertake a meticulous examination of each thematic dimension.

**Table 6.** The themes and subthemes of Fintech measures and bank profitability determinants.

| No. | Authors | Fintech Measures (Bank Level) | | | | | | | Fintech Measures (Country Level) | | | | | | Profitability Determinants | | | | | | | | | |
| | | | | | | | | | Fintech Firms | | | | | ICT | Bank-Specific Variables | | | | | | | Country-Specific Variables | | |
| | | DIGI | MT | ATR | MB | EP | FTS | IT | P2P | TPP | SHA | CRO | CC | SZ | LEV | CIR | CAR | NPL | EQ | LLP | LIQ | CON | GDP | INF |
|---|---|---|---|---|---|---|---|---|---|---|---|---|---|---|---|---|---|---|---|---|---|---|---|---|
| 1 | (Han et al. 2019) | | | | | | | | | | ✔ | | | | ✔ | | | | | | | | | |
| 2 | (Chen et al. 2020) | | | | | | | | ✔ | ✔ | | | | ✔ | | | | | | | | | | |
| 3 | (Dong et al. 2020) | | | | | | | | ✔ | ✔ | | ✔ | | ✔ | | | | | ✔ | | | ✔ | | |
| 4 | (Forcadell et al. 2020) | ✔ | | | | | | | | | | | | ✔ | | | | ✔ | | | | | | |
| 5 | (Wadesango and Magaya 2020) | ✔ | | | | | | | | | | | | | | | | | | | | | | |
| 6 | (Phan et al. 2020) | | | | | | | | | | | ✔ | | ✔ | | ✔ | | | | ✔ | | | ✔ | ✔ |
| 7 | (Tobing and Wijaya 2020) | | | | | | | | ✔ | ✔ | | | | ✔ | | | | | | | | | | |
| 8 | (Awwad 2021) | | | | | ✔ | | | | | | | | | | | | | | | | | | |
| 9 | (Almulla and Aljughaiman 2021) | | | | | | ✔ | | ✔ | | | ✔ | | ✔ | | | | | | ✔ | | | | |
| 10 | (X. Wang et al. 2021) | | | | | | | ✔ | | | | | | | ✔ | | | | | | | ✔ | ✔ | |
| 11 | (Del Gaudio et al. 2021) | | | ✔ | | | | | | | | | | | | | | ✔ | | | | | | |
| 12 | (Nguyen et al. 2021) | | | | | | | | ✔ | | | | | | | ✔ | | ✔ | | | ✔ | ✔ | ✔ | ✔ |
| 13 | (Wu and Yuan 2021) | | | | | | | | | ✔ | | | | | | ✔ | | | | | | | | |
| 14 | (Chhaidar et al. 2022) | ✔ | | | | | | | | | | | | ✔ | | | | ✔ | | | ✔ | | ✔ | ✔ |
| 15 | (Bashayreh and Wadi 2021) | | | ✔ | ✔ | ✔ | | | | | | | | | | | | | | | | | ✔ | |
| 16 | (Carlini et al. 2021) | | | | | | | | | ✔ | | | | ✔ | ✔ | | | | | | | | | |
| 17 | (Zhao et al. 2022) | | | | | | | | ✔ | ✔ | ✔ | ✔ | | | | | | | | | ✔ | | ✔ | ✔ |
| 18 | (Katsiampa et al. 2022) | | | | | | | | ✔ | ✔ | ✔ | ✔ | | | | | | | | ✔ | ✔ | | ✔ | ✔ |
| 19 | (Adalessossi 2023) | | ✔ | ✔ | | | ✔ | | | | | | | ✔ | | | | ✔ | ✔ | | ✔ | | ✔ | |
| 20 | (E. Li et al. 2023) | | | | | | | | ✔ | | | ✔ | | ✔ | | | | | | | | | | |
| 21 | (Al-Matari et al. 2022) | | | | | | | | ✔ | | | ✔ | | ✔ | | | | | | | | | | |
| 22 | (Lu 2022) | | | ✔ | | ✔ | ✔ | | | | | | | ✔ | | | | | | | | | | |
| 23 | (Haddad and Hornuf 2022) | | | | | | | | ✔ | ✔ | ✔ | ✔ | | | | ✔ | ✔ | | | | | | | ✔ |
| 24 | (Yudaruddin 2023) | | | | | | | | ✔ | ✔ | | | | ✔ | | | ✔ | | | | | | | ✔ |
| 25 | (Aguegboh et al. 2022) | | | | | | | | | | | | ✔ | | | | | | | | | | | ✔ |
| 26 | (Kharrat et al. 2023) | ✔ | | | ✔ | ✔ | | ✔ | | | | | | | | | | | | | | | ✔ | |
| 27 | (Ben Bouheni et al. 2023) | ✔ | | | | | | | | | | | | | | | | | | | | | ✔ | |
| 28 | (L. Li et al. 2023) | | | | | | | | | | | | ✔ | | | | | | | | | | | |
| | Total: | 5 | 1 | 4 | 2 | 4 | 3 | 2 | 11 | 9 | 4 | 8 | 2 | 13 | 3 | 4 | 2 | 5 | 2 | 3 | 5 | 2 | 9 | 8 |

### 3.1. Measuring Fintech Using Bank-Level Data

Drawing from the SLR findings, the most notable indicators for predicting Fintech were determined to be Digitalization (DIGI), ATM ratio, Electronic Payments (EPs), Fintech Services (FTSs), Mobile Banking (MB), and bank IT investments involving intangible resources. The first subtheme, digitization and digital transformation, has evolved in tandem with technological advancements. Digitization entails reshaping how companies engage with customers and conduct business by leveraging emerging technologies. In essence, digitization encourages the adoption of new technologies to create fresh revenue streams and seize new opportunities (Chhaidar et al. 2022; Forcadell et al. 2020). This subtheme has been referenced and employed in five articles. These studies have revealed that digitalization positively impacts banks' profitability, implying that increased digital engagement correlates with higher profitability levels (Chhaidar et al. 2022; Forcadell et al. 2020; Wadesango and Magaya 2020). Moreover, the study by Kharrat et al. (2023) scrutinized the dual banking system encompassing both Conventional and Islamic banks, and adopted digitalization as a Fintech metric. Their investigation unveiled a notably positive correlation between digitalization and profitability. Furthermore, Ben Bouheni et al. (2023) analysis elucidated a non-linear and significant association between digitalization and profitability within Islamic banking institutions.

Regarding the second subtheme, the ATM ratio. While the ATM is commonly regarded as a first wave of Fintech (Del Gaudio et al. 2021), it continues to be employed as a measure of Fintech in the literature. The ATM ratio is a component of the technological infrastructure assets and signifies investments in Information Technology (IT) (Del Gaudio et al. 2021). The SLR uncovered that four articles (Adalessossi 2023; Lu 2022; Del Gaudio et al. 2021; Bashayreh and Wadi 2021) had utilized the ATM ratio to indicate Fintech at the bank level. Two studies (Del Gaudio et al. 2021; Bashayreh and Wadi 2021) showed that a higher ATM ratio contributed to enhanced bank profitability, whereas the other two studies (Adalessossi 2023; Lu 2022) observed the contrary effect.

Electronic payments are the third subtheme. Numerous contemporary research endeavors have focused on electronic payment methods as a means to enhance service quality and subsequently improve the financial performance of banking institutions (Kharrat et al. 2023; Adalessossi 2023; Lu 2022; Awwad 2021). The articles above demonstrated that electronic payments significantly bolster profitability. In the context of the fourth subtheme, Fintech services were examined by three studies (Almulla and Aljughaiman 2021; Bashayreh and Wadi 2021; Lu 2022) as indicated by the SLR and found that it significantly promoted the profitability in banks.

### 3.2. Measuring Fintech Using Country-Level Data

Drawing upon the insights from Table 6, the most salient Fintech measures at the country level pertain to the Fintech Firms' count. In this research, Fintech entities were segregated into five distinct subcategories. The submeasure that was most recurrently harnessed to probe the correlation between Fintech and bank profitability was Peer-to-Peer lending (P2P), underscored in eleven articles. Subsequent to this was Third-Party Payment (TPP), cited in nine articles. The review also pinpointed the reference to Crowdfunding in eight articles. Furthermore, the subcategory of Shadow Banking was mentioned in a duo of articles, and Cloud Computing was invoked as a representation of Fintech in two distinct studies.

P2P lending falls within the broader theme of Fintech firms and is regarded as a country-level Fintech measure. Due to its streamlined framework, it represents online finance, enabling individuals and businesses to borrow and lend at relatively favorable interest rates (Chen et al. 2020). P2P lending was first introduced in the United Kingdom in 2005 (Chen et al. 2020). The growth of Internet finance, exemplified by Peer-to-Peer (P2P) lending and third-party payment systems, has been remarkable. This draws the attention of scholars who conduct empirical research, utilizing P2P as a Fintech measure to examine its influence on bank profitability. The outcomes were varied among the 11 studies

utilizing P2P lending platforms. Five studies reported that P2P lending platforms positively impacted profitability (E. Li et al. 2023; Al-Matari et al. 2022; Haddad and Hornuf 2022; Dong et al. 2020). Notably, one of these four studies focused on Islamic and conventional banks (Yudaruddin 2023) and revealed that P2P lending particularly enhanced the financial performance of Islamic banks, especially during both regular and crisis periods. Conversely, six studies indicated that P2P lending has a detrimental impact on bank profitability (Katsiampa et al. 2022; Zhao et al. 2022; Almulla and Aljughaiman 2021; Nguyen et al. 2021; Chen et al. 2020; Tobing and Wijaya 2020).

Shadow Banking is also considered a country-level measure of Fintech. The emergence of the subprime mortgage crisis in 2008 exposed the Shadow Banking system to the public eye (Han et al. 2019). Shadow Banking operates in credit generation, liquidity management, and maturity adjustment. However, due to its operations outside the traditional banking sector, intricate product structures, legal ambiguity, and inadequate oversight, the Shadow Banking system has evolved into a heightened systemic risk (Han et al. 2019). Furthermore, banks face significant competition from shadow banking primarily because shadow banking entities offer similar financial services, such as lending and borrowing, but often operate with fewer regulatory constraints (Katsiampa et al. 2022). The Fintech model has been employed in four studies within the literature, as indicated by this systematic literature review (Haddad and Hornuf 2022; Katsiampa et al. 2022; Zhao et al. 2022; Han et al. 2019).

The diffusion of Information and Communications Technology (ICT) and its impact on the banking sector have garnered significant attention (Aguegboh et al. 2022; Del Gaudio et al. 2021). It is also used to indicate Financial technology (Fintech). This concept can be quantified by assessing the number of ATMs per 100,000 adults or the total count of mobile money transactions per 1000 adults (Aguegboh et al. 2022). It can also be evaluated through the Broadband segment and by considering Internet users and smartphone users (Del Gaudio et al. 2021).

### 3.3. Other Bank Profitability Determinants

This review illuminated the factors influencing bank profitability in the Fintech era. It has divided the determinants of bank profitability into two categories: those associated with specific bank-related factors (Microeconomic variables) and those linked to specific country-related factors (Macroeconomic variables).

### 3.3.1. Bank-Specific Variables

Various bank-specific determinants may influence the interrelationship between Fintech innovations and the financial profitability of banking institutions. One of the primary subthemes within the bank-specific variables category is the "Bank Size". The size of a bank has been recognized as a determinant of bank profitability and has served as a control variable in most of the articles identified during the SLR ($n = 15$). The larger a bank is, the better it can mitigate risk by diversifying its asset base, exercising effective risk management and control, and reducing its vulnerability to risks (Dong et al. 2020). It is evident that the size of their assets directly influences the performance of commercial banks. As the magnitude of asset holdings escalates, there appears to be a concomitant enhancement in both the operational efficiency and the overarching performance metrics of commercial banking entities (Dong et al. 2020). Large banks should theoretically benefit from economies of scale, potentially resulting in cost or profitability advantages.

Moreover, the "too-big-to-fail" argument safeguards larger banks' market share and core income (Katsiampa et al. 2022). It has also been contended that larger banks enjoy access to more affordable capital, contributing to their robust profitability (Almulla and Aljughaiman 2021). The bank size has been used as a variable in thirteen studies of Fintech and bank profitability. Therefore, studies have shown that banks achieve economies of scale and thereby improve their profitability when they reach a significant size (Adalessossi 2023; Katsiampa et al. 2022; Chhaidar et al. 2022; Almulla and Aljughaiman 2021; Dong et al.

2020; Chen et al. 2020; Phan et al. 2020). In contrast, other research has suggested that banks with a significant asset base may not always enjoy the advantages of economies of scale, which may not necessarily lead to enhanced profitability (Yudaruddin 2023; Al-Matari et al. 2022; Lu 2022; Carlini et al. 2021; Tobing and Wijaya 2020; Forcadell et al. 2020).

Nonperforming Loans (NPLs) are one of the bank profitability determinants based on the bank level that the current study found in collected articles. One study (Adalessossi 2023) utilized it as a gauge of asset quality, while four studies (Chhaidar et al. 2022; Nguyen et al. 2021; Del Gaudio et al. 2021; Forcadell et al. 2020) employed it as a metric for assessing credit risk. Publications intended for a professional readership often emphasize commonly recognized accounting metrics like the Cost-to-Income ratio (CTI). The cost-to-income ratio serves as a gauge of a bank's efficiency. Empirical studies on banking efficacy have recurrently evidenced that an elevated cost-to-income ratio correlates inversely with holistic bank performance (Nguyen et al. 2021). Within this literature synthesis, such a variable is categorized as a subtheme nested within the broader domain of bank-centric determinants. All four of the articles (Haddad and Hornuf 2022; Wu and Yuan 2021; Nguyen et al. 2021; Phan et al. 2020) discovered that an elevated Cost-to-Income ratio (CTI) reduces bank profitability.

Furthermore, the analysis underscored the significance of Liquidity as a pivotal determinant of profitability, with references to its importance evident in four distinct scholarly articles (Adalessossi 2023; Chhaidar et al. 2022; Katsiampa et al. 2022; Nguyen et al. 2021). As per portfolio theory, increased risk is associated with more significant profit potential (Katsiampa et al. 2022). Consequently, loans are considered less liquid than other principal components of a bank's portfolio investments, and this reduced Liquidity corresponds to higher levels of risk (Adalessossi 2023). Additionally, the review revealed that the Loss Loan Provision (LLP) determines a bank's profitability. This factor was identified in a total of three empirical articles. All three articles (Katsiampa et al. 2022; Almulla and Aljughaiman 2021; Phan et al. 2020) utilized LLP to assess credit risk and indicated that heightened credit risk would reduce bank profitability. Also, the anticipation is that a decline in profitability is likely to occur due to the presence of nonperforming loans. Another subtheme that is also considered as a determinant of a bank's profitability is Bank Leverage. Some research indicated that maintaining a conservative leverage position boosts the profitability of banks; hence, banks ought to secure necessary funds by issuing equity shares (Carlini et al. 2021; X. Wang et al. 2021; Han et al. 2019).

Moreover, among the profitability determinants that this review found is Shareholder Equity. The shareholder-equity ratio signifies the proportion of the bank's assets that the owner has invested (Dong et al. 2020). A very low equity ratio suggests that the bank is heavily indebted, making it vulnerable to external shocks. Conversely, an excessively high equity ratio indicates that the bank needs to utilize financial leverage to grow its operations actively (Dong et al. 2020). Also, it assesses the bank's capacity to manage potential losses. Greater capital reduces the requirement for financing, resulting in lower risk costs due to this safety buffer (Adalessossi 2023). This particular subtheme appears twice within the collected articles, as well (Adalessossi 2023; Dong et al. 2020). Another subtheme within the category of bank-specific variables pertains to the Capital Adequacy Ratio (CAR). Capital requirements are also crucial in the strategic competition among banks (Haddad and Hornuf 2022). To remain competitive, banks must maintain a surplus of capital to prevent incurring expenses linked to regulatory intervention in cases where they come close to or dip below the mandated minimum capital ratio (Malovaná and Ehrenbergerová 2022). Drawing from the review, two scholarly articles employed this variable to investigate the nexus between Fintech and banking profitability (Yudaruddin 2023; Haddad and Hornuf 2022).

### 3.3.2. Country-Specific Variables

We comprehend that the stability of the banking sector plays a vital role in averting substantial disturbances in credit markets. Therefore, the notion of a dependable banking

system relies on the stability of macroeconomic factors like GDP and Inflation, as well as the health of the banking industry itself (Nguyen et al. 2021). This review provides insights into both industry and macroeconomic variables. Beginning with industry-related factors, we have identified that Bank Concentration can influence a bank's profitability. In accordance with the Structure–Conduct–Performance (SCP) theory, banks operating within highly concentrated markets are more inclined to engage in collusion, leading to the potential for them to attain monopoly-level profits (X. Wang et al. 2021). Furthermore, banks possessing market power have the potential to generate greater profits (Nguyen et al. 2021).

During economic upswings, lending demand typically increases, while it diminishes in downswings (Phan et al. 2020). Additionally, heightened economic growth prompts banks to issue more loans and allows for higher interest rates, thereby improving the quality of their asset portfolios (X. Wang et al. 2021). Our review identifies nine studies that highlight a significant relationship between GDP and bank profitability in various contexts, including Indonesia (Phan et al. 2020), Jordan (Bashayreh and Wadi 2021), China (Zhao et al. 2022), and in cross-country analyses (Ben Bouheni et al. 2023; Kharrat et al. 2023; Adalessossi 2023; Chhaidar et al. 2022; Nguyen et al. 2021; X. Wang et al. 2021). This review establishes that GDP is a pivotal macroeconomic factor influencing bank profitability, with its fluctuations mirroring the business cycle. In economic downturns or recessions, the quality of loan portfolios often deteriorates, leading to credit losses and reduced bank profits. Moreover, bank profits tend to synchronize with the economic cycle, primarily due to impacts on net interest income from lending activities (Phan et al. 2020).

Another subtheme, rooted in the macroeconomic context, is Inflation. The influence of Inflation on banking earnings hinges on the rate at which Inflation escalates relative to wages and other operational expenditures (Haddad and Hornuf 2022). Situations exhibiting a positive association between Inflation and profits imply that unanticipated inflationary surges, unaccompanied by commensurate interest rate adjustments by banks, can lead to a more precipitous growth in expenses compared to revenues, posing risks to banking profitability (Phan et al. 2020). These discussions suggest that the effect of Inflation on profits is uncertain (Phan et al. 2020). Based on the findings of the review, nine studies (Yudaruddin 2023; Ben Bouheni et al. 2023; Aguegboh et al. 2022; Haddad and Hornuf 2022; Zhao et al. 2022; Chhaidar et al. 2022; Nguyen et al. 2021; Phan et al. 2020) have investigated the impact of Inflation on bank profitability. Five investigations have revealed a substantial positive impact of inflation on banks' profitability (Ben Bouheni et al. 2023; Haddad and Hornuf 2022; Zhao et al. 2022; Nguyen et al. 2021; Phan et al. 2020). On the other hand, one study (Chhaidar et al. 2022) has shown that Inflation significantly negatively influences bank profitability. Additionally, one study (Yudaruddin 2023) has concluded that Inflation does not significantly impact banks' profitability. Also, one study (Aguegboh et al. 2022) showed a mixed effect.

## 4. Recommendations for Future Research Endeavors

A wide range of theories, theoretical models, and constructs were assessed across the 28 articles, offering a comprehensive overview of various Fintech measures and their influence on bank profitability, alongside other factors determining profitability, over the past five years. This review is anticipated to provide direction for future research. Recommendations for future research were formulated using the TCCM framework (Chen et al. 2021), taking into account criteria from earlier studies.

### 4.1. Theory

The examination of Fintech and its impact on bank profitability involved the utilization of various theories. These theories encompassed Consumer Theory, Disruptive Innovation Theory, Solow's Paradox Theory, Agency Theory, Financial Innovation Theory, Structure–Conduct–Performance Theory, and Transaction Cost Theory. Moreover, the most commonly investigated concepts within these studies were derived from Consumer Theory and

Disruptive Innovation Theory. While not all aspects of these renowned theories consistently proved significant, they did demonstrate predictive power under specific circumstances.

Fintech companies, such as P2P and TPP, are crucial indicators of Fintech adoption on a country level, as suggested by prior research. This underscores the significance of the Disruptive Innovation Theory and Consumer Theory as predominant frameworks. In line with the Disruptive Innovation Theory, emerging players employing innovative technology to offer more accessible and cost-effective products and services can introduce competition into the market (Phan et al. 2020). Additionally, Consumer Theory, as demonstrated in previous studies, elucidates the connection between Fintech and the profitability of banks, indicating that new services, similar to those provided by Fintech firms, can replace traditional services in response to evolving consumer demands (Almulla and Aljughaiman 2021). Agency costs arise from contractual relationships, whether they involve employees, suppliers, or customers. In this context, new technologies like keystroke monitoring, logging, and Internet surveillance have emerged as tools to monitor employees effectively, reduce agency costs, and streamline supervision (Chhaidar et al. 2022). The agency problem arises from the separation of ownership and control, which is particularly prevalent in companies with widely dispersed ownership. This misalignment of interests can ultimately result in information asymmetry and the associated agency costs. Therefore, it is recommended that governance mechanisms be established to minimize these agency costs and promote alignment of interests. Consequently, future research should focus on identifying and addressing these issues and their impact on bank profitability.

In line with Transaction Cost Theory, achieving maximum profits entails a focus on minimizing transaction costs (Chhaidar et al. 2022). To decrease these costs, organizations should adopt an optimal organizational structure, which could involve actions like pursuing forward integration or acquiring valuable skills, for instance (Chhaidar et al. 2022). With the exception of major banks, many financial institutions continue to provide traditional, expensive, and burdensome financial services. This situation opens the door for future research to explore the challenges posed by the high costs associated with outdated banking services. Future studies should also emphasize the importance of developing their own Fintech strategies or forging partnerships with Fintech firms. To accomplish this, upcoming research can enhance its empirical analysis by applying the Transaction Cost Theory. On the other hand, the Transaction Cost Theory may guide future research to delve deeper into the costs associated with banks implementing Fintech, especially in relation to the risks that banks might face when adopting their own Fintech products which may affect their profitability performance, as well.

*4.2. Context*

As per the findings from the SLR, the majority of research conducted on the relationship between Fintech and the profitability of banks has predominantly centered around conventional banks. Only a limited number of studies have delved into the influence of Fintech on the profitability of Islamic Banks (IBs) when compared to their Conventional Bank (CB) counterparts (Adalessossi 2023; X. Wang et al. 2021; Almulla and Aljughaiman 2021). Recognizing the substantial role that Islamic banks play in contributing to economic growth, there is a clear need for further future research aimed at comprehending the unique characteristics of Islamic banks. This is particularly essential due to the coexistence of IBs and CBs within the same market, even though IBs are required to tailor their banking products according to distinct principles (Sharia law), especially in the context of the rapidly expanding Islamic Fintech sector. While there have been some studies on Fintech and Islamic banks, there is a necessity for more extensive research on Islamic Fintech. Islamic Fintech has gained significant attention from practitioners and advocates of Islamic finance, particularly Sharia scholars (Chong 2021). This presents an opportunity for future research, particularly in exploring the application of blockchain technology in Islamic Fintech in alignment with Islamic law.

Furthermore, our analysis unveiled that the majority of research into the impact of Fintech and bank profitability has been conducted on individual countries, particularly in emerging Asian and Middle East countries such as China, Malaysia, Indonesia, Jordan, Palestine, and Saudi Arabia, with limited emphasis on developed markets like the United States. Among these studies, a notable portion (12 studies) has adopted a cross-country approach. Consequently, there is a need for additional research in developing countries, particularly those with substantial populations such as Egypt and Iraq, in order to uncover the challenges faced by their banks. Of paramount importance is the conduct of more comparative studies that encompass both developed and developing countries, as this would yield comparative insights and contribute to theory validation.

*4.3. Constructs*

In our examination of Fintech indicators at the bank level, we found that four key factors, specifically Digitalization, ATM ratio, E-payments, and Fintech services, emerged as the most commonly employed predictors of Fintech adoption. Conversely, at the country level, the primary Fintech measures revolve around Fintech firms, which are categorized into P2P, TPP, and Crowdfunding.

The digitalization of financial processes has the potential to cut costs by optimizing economies of scale while simultaneously improving the speed and security of transactions. Similarly, the ongoing COVID-19 pandemic has underscored the advantages of efficient financial systems in safeguarding people during times of social distancing, reduced demand, and disrupted supply chains (Chhaidar et al. 2022). Therefore, it is crucial to undertake additional research regarding the relationship between Fintech and profitability in the context of the COVID-19 pandemic. The ATM variable has been employed at both the bank and country levels. The ATM ratio represents the number of ATMs divided by the number of branches (Adalessossi 2023; Del Gaudio et al. 2021). Despite being regarded as an initial Fintech wave, ATMs continue to exert a substantial influence on bank profitability. Consequently, future research demands more exploration into using ATMs as a means to gauge Fintech's effect on bank profitability, particularly by considering ATM transaction volumes. According to the P2P, TPP, and Crowdfunding, most of the studies used them to measure Fintech and they were all considered as Fintech firms (Chen et al. 2020; Tobing and Wijaya 2020). P2P lending offers both individuals and businesses access to loans that may have been otherwise unattainable. While the third-party payment refers to a financial transaction in which a third party, separate from the bank and customer, facilitates and processes the payment (Chen et al. 2020). The influence of internet finance, specifically P2P and TPP, on bank performance has garnered significant interest from scholars, particularly concerning their direct relationship. Consequently, future research should explore the indirect impact of internet finance on bank profitability, focusing on the competitive dynamics between them. This is especially important in cases where competition is fierce, and Fintech firms pose challenges to traditional bank.

As for the other factors influencing profitability, the results of the SLR indicate that the size of the bank, the presence of nonperforming loans, and Liquidity are the most significant determinants of profitability. Bigger banks have a higher potential for risk diversification through the broadening of their asset base, which leads to more effective risk management and a decrease in their vulnerability to risks. Furthermore, it is worth emphasizing that the performance of commercial banks generally enhances as their asset size grows (Dong et al. 2020). Bigger banks derive greater advantages from investments in financial technology to enhance their performance. Moreover, as a result of economies of scale, major financial institutions are expected to possess a cost or profit edge. This notion is often associated with the "too-big-to-fail" argument (Chhaidar et al. 2022). Existing literature has a deficiency in studies that primarily consider bank size as a mediating variable, with only one study employing size as a moderator variable to explore the effects of Fintech on bank profitability (Chhaidar et al. 2022). Therefore, the importance of bank size underscores the need for

further research to investigate how bank size, either as a mediating or moderating factor, can elucidate the connection between Fintech and bank profitability.

When it comes to Nonperforming Loans (NPLs), they serve as an alternative indicator of bank stability that specifically addresses credit risk by evaluating the quality of a bank's loan portfolios (Del Gaudio et al. 2021). Previous research has consistently shown that banks facing significant credit risk tend to experience diminished profitability (Chhaidar et al. 2022; Nguyen et al. 2021). Notably, this study reveals that only one prior research (Katsiampa et al. 2022) has considered the significance of NPLs in assessing the performance of Fintech firms. Therefore, there is a need for more extensive research to delve into the influence of Fintech on bank profitability with a particular focus on the term "nonperforming loans". Furthermore, regarding the credit policy of traditional banks, strict banking regulations, especially in credit policies, can pose significant challenges for these banks. These regulations are typically implemented to enhance financial stability, safeguard consumers, and manage risks in the banking industry. While they play a crucial role in achieving these goals, they can also bring disadvantages for established banks while potentially opening doors for new players (Fintech startups) in the market (Nguyen et al. 2021). In terms of bank liquidity, Fintech companies have impacted deposit stability in several ways. Initially, deposits from individuals and non-financial businesses have shifted to interbank deposits among non-financial institutions. Secondly, money market funds are gaining greater prominence, leading to increased liquidity risk in debt. Consequently, commercial banks have had to adjust their asset composition and decrease Liquidity in response to the rapid growth of Fintech (Dong et al. 2020). In the available literature, Dong et al. (2020) are the sole source that presents evidence suggesting that the growth of Internet finance has been advantageous for commercial banks by enhancing profitability while diminishing bank liquidity. Consequently, additional research is required to provide further insights into this phenomenon.

Regarding macroeconomic indicators, this study has identified that GDP and Inflation play pivotal roles in influencing bank profitability. Both of these factors are of paramount importance to the profitability of banks and can exert a substantial impact, as indicated in studies by Adalessossi (2023) and Zhao et al. (2022). Inflation, in particular, is shown to significantly affect bank profitability. These variables are commonly considered in a majority of studies. On the other hand, bank concentration, as a determinant of profitability, appears in the literature twice (Nguyen et al. 2021; X. Wang et al. 2021). Hence, there is an opportunity for upcoming research to explore extra macroeconomic elements, notably the interest rate. This is significant, considering its challenges, importance, and its impact on bank profitability, especially in the current era of global Inflation affecting all economies. Furthermore, it would be particularly intriguing to explore the implications of interest rates as a macroeconomic factor in future research by shedding the light on the case of the Silicon Valley collapse. Where Silicon Valley Bank (SVB) experienced a major downturn resulting in the loss of billions in deposits and investments, provides a compelling subject of study. This crisis occurred against the backdrop of a flourishing tech industry during the COVID-19 pandemic, coinciding with significant client deposits. The devaluation of SVB's investments can be attributed to their holdings in US Treasuries and mortgage-backed securities, compounded by the Federal Reserve's decision to increase interest rates in an effort to combat Inflation (Guardian 2023).

Despite scientific evidence highlighting the importance of Fintech for profitability in terms of efficiency, competitive advantage, and sustainability, there is a notable aspect that requires greater attention–risk management. Risk management can provide a clearer understanding of investments from a broader perspective (Yao and Song 2021). Therefore, this study encourages future research to enrich the body of knowledge by emphasizing risk management as a key mechanism connecting Fintech and bank profitability.

In regards to economic and environmental interplay, future research should investigate the intersection of financial technology and environmental sustainability, exploring

how Fintech innovations can simultaneously enhance profitability and contribute to a greener economy.

*4.4. Method*

Every article assessed in this study employed quantitative methodologies to forecast the connection between Fintech and bank profitability. These approaches included the use of the Generalized Method of Moments (GMMs), whether in the form of Difference-GMM or System-GMM (Dong et al. 2020), and some studies utilized Ordinary Least Squares regression (OLS) (Almulla and Aljughaiman 2021). Consequently, this research recommends that future studies consider employing alternative estimation techniques, such as Seemingly Unrelated Regressions (SURs) or Quantile regression.

Seemingly Unrelated Regression (SUR) is a statistical method employed in econometrics. It serves as a tool for estimating and examining a set of regression equations in situations where the error terms across these equations might exhibit correlations (Gordon 2015). SUR permits researchers to concurrently estimate multiple regression equations, which may be interconnected but not identical. This approach finds frequent application in a range of disciplines, notably in economics and finance, where researchers aim to scrutinize how various dependent variables are affected by a shared set of independent variables, all while taking into account potential error term correlations (Gordon 2015). While Quantile regression is a statistical method utilized in econometrics and data analysis for examining the connection between variables in a manner that extends beyond the conventional linear regression model (Chang et al. 2020).

It proves particularly useful when the assumptions inherent in OLS regression, such as homoscedasticity and normality of residuals, are not satisfied. This technique offers a more holistic perspective of data distribution and allows for the detection of variations in the impact of predictors across various quantiles (Chang et al. 2020).

*4.5. Key Implications for Policymakers*

This study provides policymakers with crucial insights concerning the banking sector and the potential impact of the Fintech revolution, particularly in credit policy provision. The preceding discussion in the paper indicates that stringent banking regulations, coupled with the rise in Fintech companies in the financial services sector, have played a role in this disruption. Specific Fintech solutions, such as Crowdfunding, have attracted clients away from traditional banks due to their less rigid regulations regarding the credit system. Further important consideration for policymakers involves the imperative for cross-disciplinary discussions and global collaboration to create a fundamental regulatory framework and a set of policies pertaining to Fintech. A common set of regulations is crucial to, among other objectives, maintain the stability of the financial system, ensure the seamless operation of financial markets, protect consumers, and improve social and economic welfare.

**5. Conclusions**

From the 28 articles we retrieved from the WoS and Scopus databases, it is evident that there is a burgeoning academic interest in the intersection of Fintech and bank profitability. This heightened attention to the subject has led to the creation of several influential articles that contribute advanced knowledge, establishing a solid basis for future research. These 28 studies, conducted between August 2019 and August 2023, encompass a diverse array of theoretical models and frameworks exploring the relationship between Fintech and banks' profitability. This study has also categorized Fintech measures into two groups based on bank-level factors: digitalization, ATM ratio, E-payments, money transfer, mobile banking, IT investment, and Fintech services. Additionally, at the country level, these measures encompass Fintech firms such as P2P, TPP, Crowdfunding, shadow banking, and cloud computing, as well as ICT. The research has also delineated two distinct sets of determinants affecting bank profitability: bank-specific variables (including size, Liquidity,

nonperforming loans, cost-to-income ratio, etc.) and macroeconomic variables (such as concentration, GDP, and Inflation).

In the last five years of research on Fintech and its impact on bank profitability, a total of 113 associations were uncovered, shedding light on the comprehensive landscape of Fintech measures and determinants influencing profitability. These findings serve as a foundation for forthcoming studies in this domain. Among the frequently employed measures at the bank level are digitalization, ATM ratio, and E-payment, while at the country level, P2P, TPP, and Crowdfunding are prominent choices. In terms of other determinants of profitability, this study highlights bank-specific variables such as size, Nonperforming Loans (NPLs), and Liquidity as commonly studied factors. Regarding macroeconomic variables, both GDP and Inflation emerge as frequently explored elements in the existing literature.

**Author Contributions:** Conceptualization, A.T., A.A.-R., S.I.M.A. and M.F.G.; methodology, A.T., A.A.-R., S.I.M.A. and M.F.G.; software, A.T., A.A.-R., S.I.M.A. and M.F.G.; validation, A.T., A.A.-R., S.I.M.A. and M.F.G.; formal analysis, A.T., A.A.-R., S.I.M.A. and M.F.G.; investigation, A.T., A.A.-R., S.I.M.A. and M.F.G.; resources, A.T., A.A.-R., S.I.M.A. and M.F.G.; data curation, A.T., A.A.-R., S.I.M.A. and M.F.G.; writing—original draft preparation, A.T., A.A.-R., S.I.M.A. and M.F.G.; writing—review and editing, A.T., A.A.-R., S.I.M.A. and M.F.G.; visualization, A.T., A.A.-R., S.I.M.A. and M.F.G.; supervision, A.T., A.A.-R., S.I.M.A. and M.F.G.; project administration, A.T., A.A.-R., S.I.M.A. and M.F.G.; funding acquisition, A.T., A.A.-R., S.I.M.A. and M.F.G. All authors have read and agreed to the published version of the manuscript.

**Funding:** This research was funded by Yayasan Tun Ismail Ali Endowment Research Grant at UKM [(grant code YTI-UKM-2023-004)].

**Informed Consent Statement:** Not applicable.

**Data Availability Statement:** Data can be provided upon request.

**Conflicts of Interest:** The authors declare no conflicts of interest.

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
