# Peer review of "A Systematic Review of Fintech and Banking Profitability"

_ijfs, doi:10.3390/ijfs12010003_

Round 1
Reviewer 1 Report
Comments and Suggestions for Authors
The title is focussed: A Systematic Review of Fintech and Banking Profitability"
Abstract: profitability is mentioned 4 times/100
Introduction:
Background information needs more about details about profitability and factors impacting it. I think the profitability issue is not clearly synthesized in the introduction. Profitability was mentioned 6 times/100 in the introduction and the factors as per the abstract was not clearly articulated here. Unclearly conceived, established the connections and synthesized into the intro the factors that determine bank profitability.
Lines 78 to 93; 97 to 99; 106 to 107; 109 to 115; 117 to 120; 155 to158; 562 to 563: Each statement made here has not been verified or validated with any compelling in-text reference evidence. Please check through each sentence and make sure that it is not just a sweeping statement not supported by any evidence.
Line 127 to 145 is a duplication of Lines 108 to 126
knowledge gap:
"Therefore, this systematic review tries to bridge 123 these gaps and challenges through increased research and analysis is crucial for unlocking 124 the full potential of Fintech and its positive impact on innovation, economic growth, and 125 societal development particularly in the banking context"an "Therefore, this systematic review tries to bridge 142 these gaps and challenges through increased research and analysis is crucial for unlocking 143 the full potential of Fintech and its positive impact on innovation, economic growth, and 144 societal development particularly in the banking context." (be more specific around determinants of profitability as these gap statements are too broad.) What gap is your article trying to close exactly/
Methodology:
What software was used for the data text mining and to analyse the content of the collection of articles?
How were the documents examined, what criteria was used to eliminate documents.
Lines 496 to 500- Move to be after the topic sentence (Lines 496 and 497) to support the topic sentence of the paragraph lines 496 to 509.
Discussion and Results were done fairly well. Try and see if any of the discussion can be converted into an infographic to make it easier to digest. factor and sub-factors in a diagram is easier to digest than written word.
Conclusion summarises the findings, the limitations seem briefly evident. If I read the introduction and then the conclusion I am unable to see the connection clearly. So write the introduction in a way so that I do not need to read the methodology and results and discussion section to know what your ultimate message is.
Author Response
Dear reviewer,
Thank you very much for taking the time to review this manuscript. Please find the detailed responses below and the corresponding revisions/corrections highlighted.
Best regards

Reviewer 2 Report
Comments and Suggestions for Authors
I have read the paper with great interest and it is interesting and as a reviewer I have to provide constructive criticism to improve the quality of the manuscript. I invite authors to consider the following comments to improve the quality of their work.
1-The study begins with a broad introduction to fintech but does not clearly state its research objectives or questions. It's important for readers to know what specific aspects of fintech and bank profitability the study aims to address.
2-The study claims to offer a systematic review but restricts its analysis to only 28 articles published between August 2019 and August 2023. Given the rapidly evolving nature of fintech, this narrow scope may not provide a comprehensive overview of the topic. Authors can develop this aspect and add more important modern works.
3-The study mentions adhering to the PRISMA guidelines, but it lacks transparency regarding the systematic review methodology. Details about the inclusion and exclusion criteria for the selected articles, as well as the data extraction process, are essential for evaluating the quality of the review. This study can benefit from and cite the following literature to enhance this aspect:
https://doi.org/10.3390/ijfs11030092
4- In general, the review summarizes findings from the selected articles but lacks critical analysis and synthesis of the results.
5-The study claims to introduce a "holistic methodology," but it primarily focuses on quantitatively assessing fintech measures and their impact on bank profitability. A more holistic approach should also consider qualitative aspects, such as user experiences and regulatory challenges. I invite the authors to justify or dismiss this point.
6-While the study mentions offering insights for "practical implementation," it does not adequately discuss the real-world implications of its findings. How can banking professionals and policymakers use this information to make informed decisions?
7- The study mentions charting the course for future research but does not provide specific recommendations or directions for future studies in the field of fintech and bank profitability. This is a missed opportunity to guide further research.
8-The study contains some grammatical issues and awkward phrasing that need to be addressed to improve the overall readability and professionalism of the paper.
Comments on the Quality of English Language
The study contains some grammatical issues and awkward phrasing that need to be addressed to improve the overall readability and professionalism of the paper.
Author Response

(The authors gave the same response as above.)
